# Ordering of room-temperature magnetic skyrmions in a polar van der Waals magnet

Peter Meisenheimer [1,8] ✉, Hongrui Zhang [1,8] ✉, David Raftrey [2,3], Xiang Chen [2,4], Yu-Tsun Shao[5], Ying-Ting Chan[6], Reed Yalisove[1], Rui Chen[1], Jie Yao [1], Mary C. Scott [1,7], Weida Wu [6], David A. Muller [5], Peter Fischer [2,3], Robert J. Birgeneau [2,4] & Ramamoorthy Ramesh [1,2,4]

Control and understanding of ensembles of skyrmions is important for realization of future technologies. In particular, the order-disorder transition associated with the 2D lattice of magnetic skyrmions can have significant implications for transport and other dynamic functionalities. To date, skyrmion ensembles have been primarily studied in bulk crystals, or as isolated skyrmions in thin film devices. Here, we investigate the condensation of the skyrmion phase at room temperature and zero field in a polar, van der Waals magnet. We demonstrate that we can engineer an ordered skyrmion crystal through structural confinement on the μm scale, showing control over this order-disorder transition on scales relevant for device applications.

Skyrmions are topologically nontrivial quasiparticles made up of a collection of rotating magnetic spins[1-3] or, more recently, electric dipoles[4,5]. In many magnetic systems, this stems from broken inversion symmetry in the crystal, resulting in a Dzyaloshinskii-Moriya interaction (DMI), or at interfaces, leading to the Rashba effect, that can stabilize such topologically nontrivial spin configurations[2,6]. Due to the topological protection from weak fluctuations and the ability to manipulate them with a magnetic field or charge current, magnetic skyrmions are of significant interest for next generation information technologies[7-11]. Neuromorphic computing architectures based on skyrmions have been proposed, using the diffusion of skyrmions and the resulting magnetoresistance to carry out probabilistic computing[12-14]. Additionally, magnetic skyrmions can be used to control spins through the topological Hall effect, where an electron interacts with the Berry curvature of the skyrmion to generate an emergent electric field[15-17]. These phenomena, both skyrmion motion and the topological Hall effect, are sensitive to the ground state of the system, wherein the arrangement of the skyrmions can have significant consequences on the resulting properties[6,18-20]. A fundamental understanding and pathways to control such ensembles of skyrmions is then important for realization of advanced computational devices

based on skyrmion kinetics or spin transport. In this vein, the arrangement of magnetic skyrmions becomes exciting from perspectives of both fundamental physics and device engineering.

Since their discovery, control of the shape[21-24] and long range ordering[25-27] of magnetic skyrmion ensembles has been an open challenge, often explored through magnetic field, temperature and thin film interactions. In two-dimensional (2D) systems, in the absence of structural defects or kinetic limitations, the behavior of such 2D ensembles can be described by the Hohenberg-Mermin-Wagner theorem, which states that true long-range, crystalline order cannot exist due to logarithmic fluctuations of the order parameter. This occurs because the energy associated with site displacement is small enough in 2D that the increase in entropy associated with the formation of density fluctuations becomes favorable[28-30]. An extension of this is the Kosterlitz–Thouless–Halperin–Nelson–Young (KTHNY) theory, which describes the melting of 2D objects with continuous symmetry through the unbinding of topological defects due to thermal fluctuations[31-33]. KTHNY melting is distinct from 3D phase transitions in that it proceeds from an ordered solid to a disordered liquid through a quasi-ordered hexatic phase that is present in 2D systems. The solid-to-hexatic phase transition is driven by the dissociation of dislocation

[1]Department of Materials Science and Engineering, University of California, Berkeley, CA, USA. [2]Materials Sciences Division, Lawrence Berkeley National Laboratory, Berkeley, CA, USA. [3]Department of Physics, University of California, Santa Cruz, CA, USA. [4]Department of Physics, University of California, Berkeley, CA, USA. [5]School of Applied and Engineering Physics, Cornell University, Ithaca, NY, USA. [6]Department of Physics, Rutgers University, New Brunswick, NJ, USA. [7]Molecular Foundry, Lawrence Berkeley National Laboratory, Berkeley, CA, USA. [8]These authors contributed equally: Peter Meisenheimer, Hongrui Zhang. ✉e-mail: meisep@berkeley.edu; hongruizhang@berkeley.edu

pairs; when dislocations with opposite Burgers' vectors are coupled, as in the 2D solid phase, the topology of the lattice of skyrmions remains invariant and order is preserved. Upon dissociation of the dislocations, long-range translational order is destroyed, but orientational order is preserved. The hexatic-to-liquid phase transition is then driven by thermal unbinding of these dislocations into disclinations, which destroys any remaining orientational order[25,31-35]. This quasi-long-range degree of crystallinity in the hexatic phase distinguishes it from the liquid phase quantitatively, as the orientational correlation function decays algebraically instead of exponentially (Supplementary Fig. 1).

As a 2D arrangement of quasi-particles, one could envision that the ensemble of magnetic skyrmions can be explained within the framework of the KTHNY theory. This has been observed in certain cases[25], as well as in other 2D systems such as colloids, liquid crystals, atomic monolayers, and superconducting vortices[34-39]. The presence of crystallographic and/or magnetic disorder, however, will interfere with the ideal KTHNY behavior. Pinning sites that can affect the thermal motion and (dis)association of dislocations and disclinations will prevent model KTHNY phase transitions, and subsequently limit ordering to short-range[40,41]. In this way, phase transitions will be suggestive of hexatic and ordered phases, but the ordered state may not be realized on experimental timescales[27,42].

It has been previously shown, using Lorentz transmission electron microscopy, that Bloch-type magnetic skyrmions in bulk, single crystal $Cu_2OSeO_3$ can undergo KTHNY-type melting with a magnetic field[25]. Additionally, skyrmion ordering has been studied in the ultrathin magnetic film Ta (5 nm)/$Co_{20}Fe_{60}B_{20}$ (0.9 nm)/Ta (0.08 nm)/MgO (2 nm)/Ta (5 nm), but KTHNY behavior was only alluded to and not fully realized[27]. The authors claim that the increased size of orientational domains in the sample is indicative of a transition to the hexatic phase, but the timescales required for the system to fully relax kinetically are not reasonably achievable. With this as background, in this work, we study the formation and solidification of Néel-type magnetic skyrmions in a novel layered, polar magnetic metal, $(Fe_{0.5}Co_{0.5})_5GeTe_2$.

Here, $(Fe_{0.5}Co_{0.5})_5GeTe_2$ (FCGT) is studied as a polar magnetic metal, a metallic magnet which intrinsically breaks spatial inversion symmetry, that hosts stable magnetic Néel skyrmions at room temperature and zero magnetic field[43,44] (Supplementary Fig. 2). The parent material $Fe_5GeTe_2$ is a centrosymmetric ferromagnet[45,46], but breaking of spatial inversion symmetry emerges at 50% Co alloying, potentially due to chemical ordering of the transition metal site and a switch from ABC to AA' stacking of the layers[44] (Fig. 1a). Broken inversion symmetry gives rise to a bulk Dzyalozhinski-Moriya interaction (DMI), allowing for stabilization of stripe cycloidal domains and/or a Néel skyrmion phase at room temperature[2]. Both the stripe domains and skyrmions can be observed using magnetic force microscopy (MFM), which probes the magnetostatic force on a scanning probe tip. Here, we use this technique to study the quasi-static phase changes of the skyrmion ensemble in an exfoliated sample as a function of temperature and to evaluate our control over their ordering (Fig. 1b, c).

Understanding the emergence of long range order in the FCGT skyrmion ensemble is the focus of this work. We observe an exponential change in the order parameters through the phase transition, suggesting that an ordered phase may be achievable at room temperature, if the role of structural/chemical defects or kinetic obstacles can be overcome. Through spatial confinement of the device which imposes an ordering director field, we are able to overcome this pinning energy and engineer an ordered skyrmion crystal at room temperature, demonstrating control over this order-disorder transition on scales relevant for device applications.

## Results

In a 125 nm thick FCGT flake, when slowly cooled below the stabilization temperature of ~314 K, the helimagnetic domains present at high

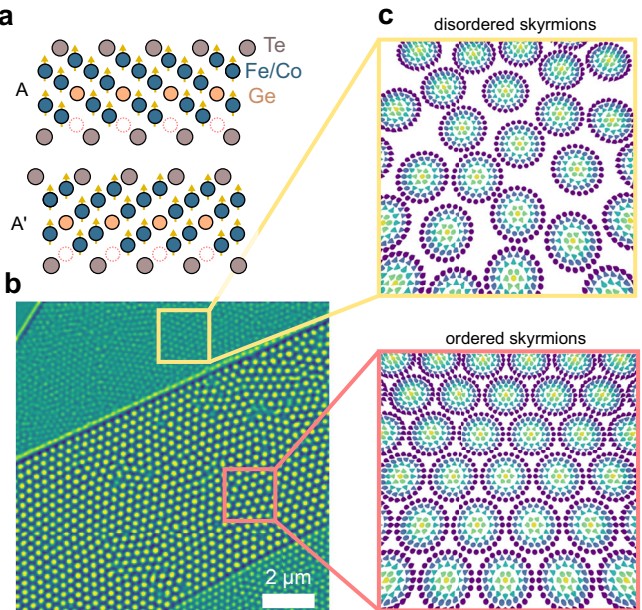

**Fig. 1 | Layered polar magnet. a** Illustration of the crystallography of FCGT, showing the AA' stacking which breaks inversion symmetry and allows for a non-zero DMI. **b** Sample with Néel skyrmions showing multiple regimes of, (**c**), ordered (close packed, bottom) and disordered (top) skyrmion configurations.

temperature condense into an assembly of Néel skyrmions[44], when perturbed by a magnetic field. This stabilization temperature is significantly below the $T_c$ of 350 K. The assembly of skrymions is initially disordered, but its correlation length progressively increases as the temperature is further decreased, reaching a stable, but still disordered, state at ~306 K (Fig. 2a). It can be reversed with temperature, melting from the stable glassy state and increasing in disorder, until the skyrmions disappear into a uniform magnetic state (Supplementary Fig. 4). In the 2D-solid phase, lattice sites should be largely 6-fold coordinated, disturbed only by pairs of correlated dislocations (neighboring pairs of neighboring 5- and 7-fold sites) with opposite Burgers' vectors, preserving translational and orientational order[31-33]. In the hexatic phase, these dislocations should no longer be bound, and free dislocations allow only quasi-long range orientational order. In our system, where 5(7)-fold coordinated sites are shown in purple (green) in Fig. 2a, b, free dislocations are still present in significant concentrations below the transition temperature at ~306 K, indicating that we are not reaching a 2D-solid phase. Additionally, we see an increase in the number of free 5- or 7-fold sites (disclinations), as we increase the temperature from 300 K to 314 K. This is a hallmark of a 2D-liquid, as the presence of disclinations will destroy the quasi-long-range order. These phase changes are generally suggestive of KTHNY behavior, but the fact that we have never observed the 2D-solid phase may be indicative of magnetic or crystalline defects randomly perturbing the system, forcing the skyrmion lattice into a state of quenched disorder when measured at laboratory timescales[27]. In the low-temperature, frozen state, ~75% of sites are 6-fold coordinated, which decreases exponentially as temperature is increased, reaching ~40% of sites being 6-fold just before the complete dissolution of the skyrmion phase (Fig. 2d, e). This exponential trend is mirrored in the correlation length extracted from the radial distribution function, which increases from melting to ~308 K, at which point it saturates to a finite value. This saturation of the exponential correlation length is again indicative of frozen translational disorder, as this parameter should otherwise diverge in a solid[47].

Quantitative assignment of phases is possible through the bond orientational parameter, which exhibits different behaviors in the

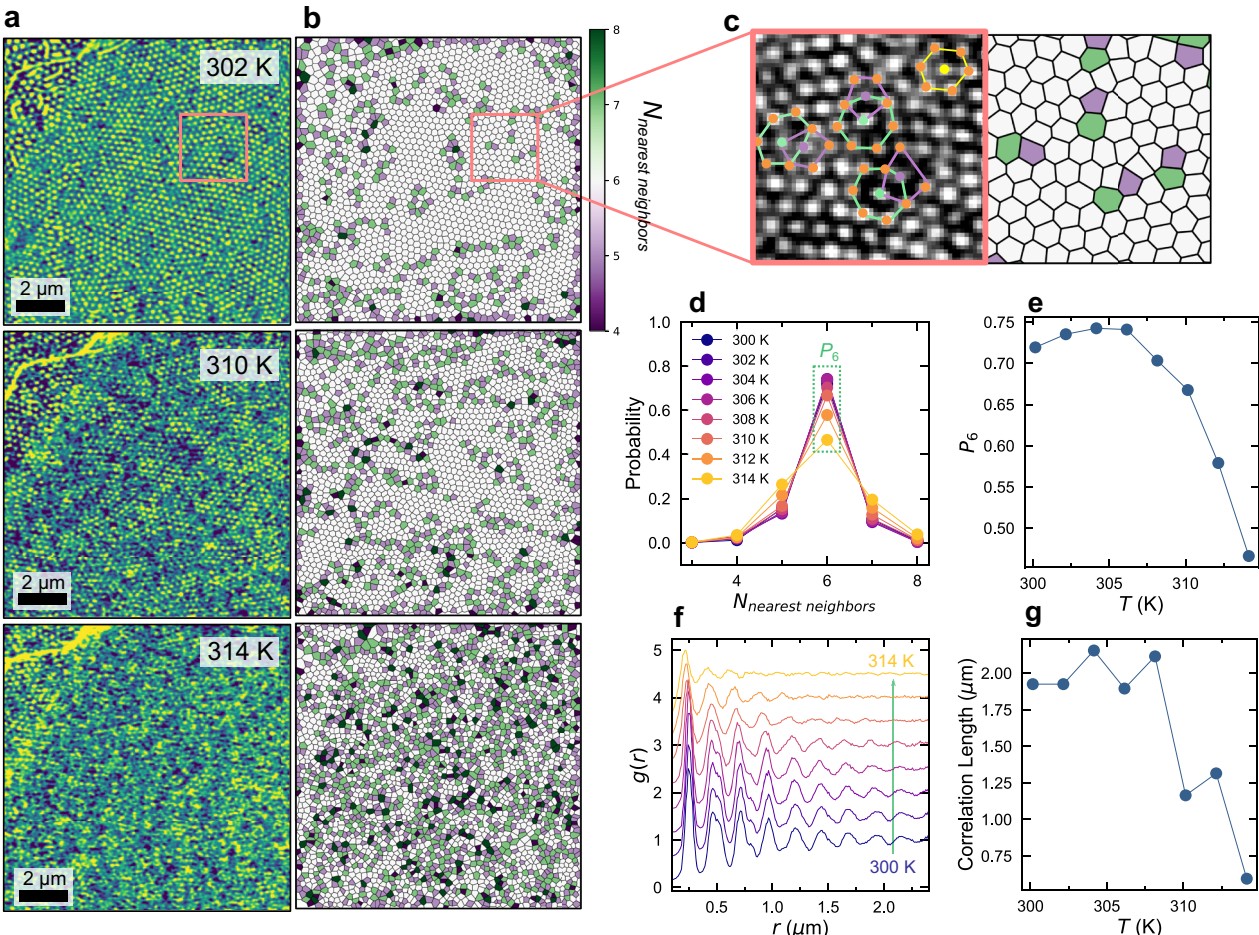

**Fig. 2 | Temperature dependence of topological defects. a** MFM images of the FCGT skyrmion lattice at different temperatures. Scale bars are 2 µm. **b** Voronoi polyhedra showing the number of nearest neighbors for each skyrmion in the image. Green(purple) sites have more(less) than 6 nearest neighbors. When the green and the purple sites neighbor one another, a dislocation is formed from bound topological defects, which is indicative of the disordering in the hexatic phase. **c** Magnified image of bound defects, illustrating the 5–7 pairs caused by displacement of a single line of skyrmions. The image is scaled up from the box in (a

and **b**). **d** Histogram of the number of nearest neighbors per site per temperature. We see that that the variance of the Gaussian distribution increases as the temperature is increased. **e** Peak value of the probabilities of an ideal 6-fold site in (**d**), showing exponential decay above ~306 K. **f** Radial distribution function and, (**g**), correlation length as a function of temperature, showing similar decay as temperature is increased. Correlation length is defined as the last peak in (**f**) before the intensity decays to 5% of the initial peak.

solid, hexatic, and liquid phases[25]. The bond orientational parameter can be defined as:

$$\Psi_6(\mathbf{r}) = \frac{1}{N_{nn}} \sum_j^{N_{nn}} e^{-6i\theta_j},  \quad (1)$$

where $N_{nn}$ is the number of nearest neighbors for the site at position $\mathbf{r}$ and $\theta_j$ is the angle between the bond to nearest neighbor $j$ and an arbitrary, but fixed, axis. This asymptotes to 1 if the bonds to the nearest neighbor sites are arranged in a hexagonal array, and to 0 if far from hexagonality. This is shown in Fig. 3a, where the shade of blue indicates the value of $|\Psi_6(\mathbf{r})|$. The correlation function of $\Psi_6(\mathbf{r})$,

$$g_6(r) = \frac{1}{N_r} \sum_{i,j}^{N_r} \Psi_6(\mathbf{r}_i) \Psi_6^*(\mathbf{r}_j),  \quad (2)$$

where $N_r$ is the number of sites distance $r$ apart, can then be used to quantitatively define the phase of the system. A solid phase will have long range order and $g_6(\mathbf{r})$ will not decay over distance, the hexatic phase will have short- but not long-range order and $g_6(\mathbf{r})$ will decay algebraically, and in a liquid phase will decay exponentially (Supplementary Fig. 1). Here, we observe exponential decay of $g_6(\mathbf{r})$ at all

temperatures (Fig. 3). This indicates that our system is in the 2D liquid phase at all temperatures. The rate of decay, $\xi$, however, follows the same exponential trend with temperature as the probability of nearest neighbors in Fig. 2, implying a phase change at the saturation temperature. We hypothesize that this condensation process between ~314 and 306 K, which seems to kinetically freeze the skyrmions into a glassy state with very-short-range ordered domains, would be the liquid-to-hexatic phase transition described in KTHNY theory but is inhibited by kinetic pinning. This is likely due to crystallographic defects which introduce inhomogeneities into the magnetic or orbital lattices[40,41,48,49].

In plan-view dark field (scanning) transmission electron microscope images (DF-(S)TEM), FCGT flakes show clear dislocation lines with spacings on the order of 100–200 nm (Fig. 4a). In a layered hexagonal system, these are similar to the well-studied stacking faults in graphite and other vdW crystals[50–55]. Under a 2-beam condition, g·b analysis shows that these dislocations possess Burgers' vectors along $\langle 11\bar{2}0 \rangle$, Fig. 4b, consistent with the stacking faults that most easily form in layered hexagonal systems due to the ease of in-plane slip[50,56]. We can then use 4D-STEM (Methods) to reconstruct the strain fields $\varepsilon_{11}$, $\varepsilon_{22}$, and $\varepsilon_{12}$ that occur due to these stacking faults, observing values of up to ~1.5% strain (Fig. 4d–g). Nonuniform fields of this mangitude

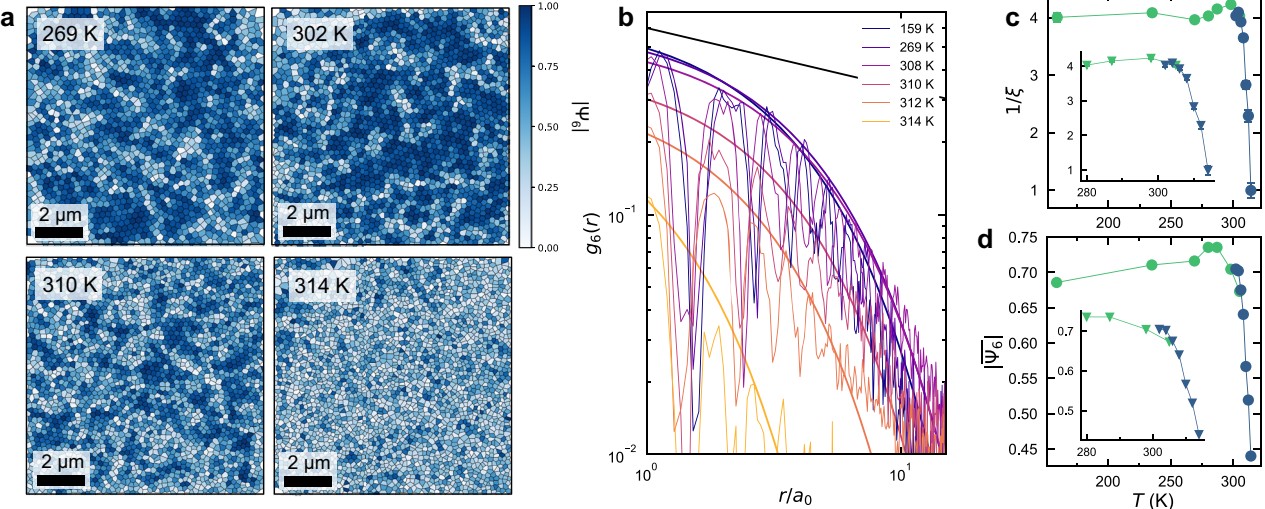

**Fig. 3 | Orientational order parameter. a** Bond orientational maps of skyrmions as a function of temperature, where the magnitude of the bond orientational parameter, $|\Psi_6(\mathbf{r})|$, is shown in blue. Qualitatively, the orientational order of the lattice decreases with increasing temperature. Scale bars are 2 µm. **b** Orientational correlation function, $g_6(r)$, as a function of temperature, which follows an exponential decay at all temperatures. Fits to $e^{-\frac{r}{\xi}}$ are shown as dotted lines. The predicted ideal KTNHY algebraic decay behavior with exponent 1/4 is shown as a solid black line. **c** The exponent $\xi$ from panel (**b**) plotted as a function of temperature, showing a clear exponential decrease as temperature is increased. Inset is the temperature range above saturation. **d** Average $|\Psi_6(\mathbf{r})|$ as a function of temperature, that shows exponential decay of the orientational order with respect to temperature falling off quickly above ~306 K. Inset shows the temperature range of clear exponential behavior. Light green points in parts (**c** and **d**) are superimposed from low-temperature MFM measurements.

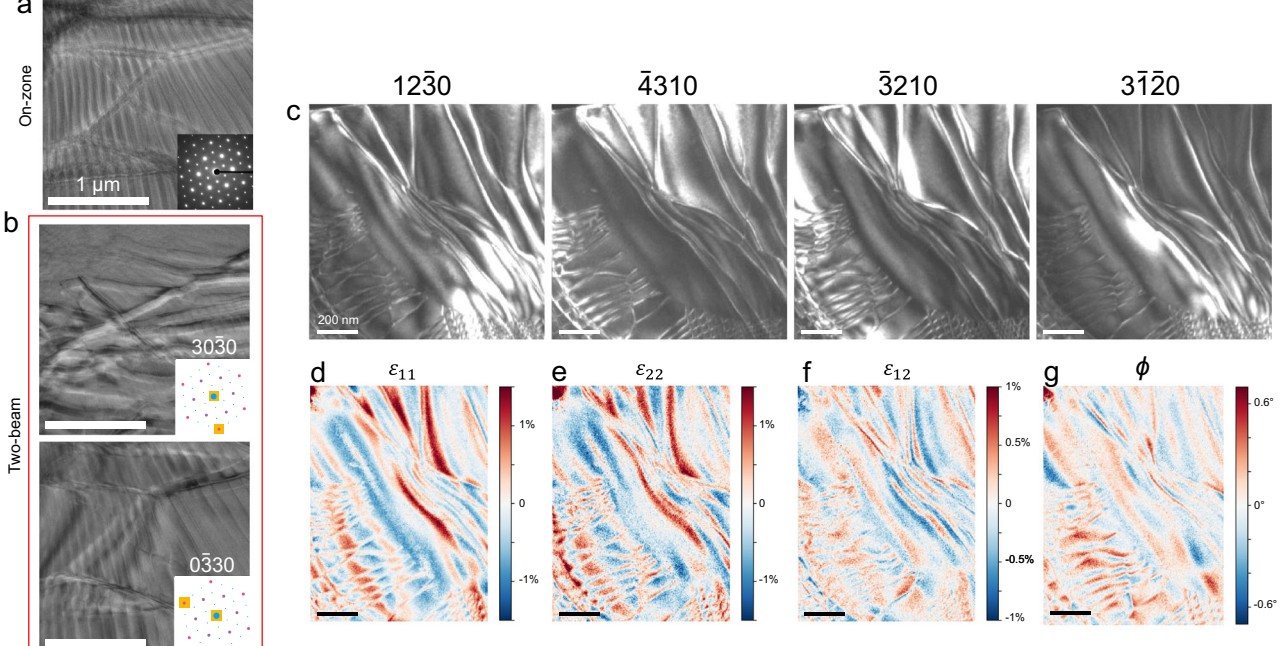

**Fig. 4 | (S)TEM imaging of defects. a** DF-TEM shows the nominally single phase flake hosts dislocations with periodicities on the order of ~100 nm. **b** In two beam conditions along the $30\bar{3}0$ and $0\bar{3}30$ axes, these dislocations decrease in contrast, indicating that the Burgers' vectors lie along $\langle 11\bar{2}0 \rangle$. **c** Virtual dark-field images can also be generated from a 4D-STEM dataset by integrating the Bragg peak intensities from reflections. From this data, we are able to calculate the projected in-plane strain fields $\varepsilon_{11}$ (**d**), $\varepsilon_{22}$ (**e**), and $\varepsilon_{12}$ (**f**), along with the in-plane rotation $\phi$ (**g**). We note the mis-tilt artifact is inevitable in the present method for strain determination, as shown in red diagonal stripes in (**d** and **e**).

would be sufficient to disrupt KTHNY behavior, indicating that these structural defects inherent to layered materials are preventing condensation of the hexatic phase[34,35,57].

　Defect-driven quenching of phase transitions and order in magnetic systems is specifically reminiscent of the breakdown of long-range order in random field Ising magnets[41,42], where magnetic defects make ordering kinetics extremely slow and the disordered state becomes frozen at low temperatures[47]. This is qualitatively consistent with previous results which show that pinning from crystalline defects can impede skyrmion motion[1,8,58]. In our study, this is supported by the

average of $\Psi_6(\mathbf{r})$, in addition to the correlation length in Fig. 2g, which saturates below ~308 K, instead of continuously increasing as is observed in typical hexatic transitions[3,25,27]. Our conclusion is also in agreement with the results in ref. 27, in which the authors also do not observe a true hexatic phase in a magnetic skyrmion lattice, indicated by exponential decay in $g_6(r)$, but attribute it to the arbitrarily long timescales associated with condensation. If this is true and condensation of the hexatic and solid phases is not achievable on a reasonable time scale due to pinning, can we engineer the introduction of a new field that favors ordering to overcome these kinetic limitations and influence formation of the ordered phase? Intuitively, this should be possible, as it is analogous to other disordered functional systems such as relaxor ferroelectrics[59] and spin glasses[60] where strong fields can force an otherwise frozen/disordered order parameter into a long-range ordered state[61].

## Discussion

Previous studies on magnetic skyrmion lattices have indicated that physical confinement may influence the formation of isolated skyrmions or skyrmion arrays[24,26,62–64]. It has been previously shown in FeGe nanowires fabricated from bulk, single crystals, that creating devices on the scale of ~4–5 skyrmion widths can force the skyrmions into a close packed array[26]. This is an important consideration for potential device-scale applications, as the skyrmion lattice will impact dynamics and transport[9]. This has, however, not been investigated when it comes to skyrmion crystal phases, and the stability of the skyrmion lattice. Here, we first simulate confinement using the boundary conditions of our micromagnetic simulations, which has previously been used to accurately describe the behavior of skyrmions in FCGT[43,44].

In our micromagnetic simulations, we artificially create a confined strip of material by fixing the left and right boundary conditions, while making the top and bottom periodic (hereafter semi-periodic) Fig. 4. Compared to a cell with fully periodic boundary conditions, we see a clear difference where the skyrmions in the periodic cell relax to a disordered, liquid ground state, while the skyrmions in the confined, semi-periodic cell relax to a hexatic phase (Fig. 5). This difference is also clearly seen qualitatively in the structure factor (inset in Fig. 5a) and in the number of dislocation sites, Fig. 5b. In this simulation, we speculate that ordered skyrmions form at the fixed boundary, which allows for nucleation of larger, more uniform domains. We observe this more directly in Fig. 5d, showing the Euler angle $\Theta = \arg(\Psi_6(\mathbf{r}))/6$. $\Theta$ indicates the average rotation of a skyrmion lattice site with respect to the $x$-axis and allows us to distinguish domains of uniform rotation/mosaicity[27]. Here, the color corresponds to the average rotation of the site, meaning areas of uniform color correspond to mosaic domains in the skyrmion lattice. With periodic boundary conditions, we see small, well-defined domains, which are largely absent in the simulated images with fixed boundaries. Additionally, looking at the average order parameter as a function of the spacing between the fixed boundary conditions, we see that in the smallest case, when the lattice is most confined, the skyrmions are most ordered, then approaching the liquid phase as the spacing is widened and the area becomes bulk-like (Supplementary Fig. 6). We assert that the presence of the boundary exerts an orientating director field, conjugate to the orientational order parameter, making true long range orientational order of the skyrmions possible. This enhances both the orientational and positional orders of the skyrmion crystal. We note, however, that in our engineered systems KTHNY theory is no longer directly applicable. KTHNY theory is true in the infinite limit, thus the confined region is explicitly not pure-KTHNY behavior. Notably, however, the liquid-hexatic-phase transition can still be observed in systems where nonlinear potentials and finite size effects are present[65–68], making the KTHNY terminology a useful framework for the interpretation of the order-disorder transition.

Experimentally, we create spatial confinement by overlapping two thick (114 nm and 214 nm) FCGT flakes in a single rectangular region. These thicknesses were chosen as they are both within the regime to stabilize Néel skyrmions as shown by previous work[44]. The flakes do not interact electrically due to a large thickness of flakes compared to the interface and the transfer process in the air, leading to ~5 nm of adsorbates at the interface. This implies their interaction is governed by long range magnetostatics and this overlapped bilayer region behaves like a thicker region within a semi-periodic, but finite, potential. This semi-classical interaction is supported by the fact that the skyrmions in the bilayer region are ~1.5x the size of those in the thinner regions (Fig. 6e), which is consistent with the thickness dependent size changes in FCGT governed by Kittel's law[44]. This is also illustrated in Supplementary Fig. 9, where the simulated bilayer hosts skyrmions that extend through the entire structure, effectively behaving as a thicker magnet. In the free monolayer and confined bilayer regions, the results parallel those of the simulation, where the monolayer region is stable in a 2D liquid phase, but the confined, bilayer region is in a hexatic phase. We consider the monolayer region to be bulk-like due to its lateral size compared to that of the skyrmions. This is quantified by the structure factor (Fig. 6a), where we see clear differentiation between amorphous-like and crystalline behaviors, and the $g_6(r)$ parameter (Fig. 6f), which decays exponentially in the monolayer region and algebraically in the bilayer region. We posit that this is due to the imposition of a uniaxial director field at the boundary, which overcomes the elastic energy of the native defects and effectively decreases the energy required to grow larger, more ordered domains. In this case, where the device size is comparable to ~20x the skyrmion lattice constant, this director field is enough to override the kinetic limitations preventing stabilization of the ordered hexatic phase. Another factor to this difference between the monolayer and bilayer samples may be that strain and pinning defects in the bilayer are averaged across the two flakes, potentially reducing the composite nonuniformity. This averaging of properties can also be observed with the equilibrium lattice constant of the skyrmions in the bilayer, where we see the anisotropies of the individual layers average out in the bilayer system and behave like a single sample.

Here we investigate the stability of the lattice of Néel-type skyrmions in FCGT. We show that the system does not follow true KTHNY behavior and condense into a 2D solid, but does undergo a freezing transition at ~304 K, significantly below the $T_c$, where the disordered skyrmions are frozen into a more ordered, but still glassy, lattice, down to cryogenic temperatures. From (S)TEM imaging, we show the presence of native dislocations in the sample which lead to large non-uniform strain fields sufficient to disrupt KTHNY behavior. We show that by confining the device, however, we are able to impose a strong enough ordering field to overcome this elastic nonuniformity, stabilizing a hexatic phase and engineering the order-disorder transition.

## Methods

### Sample synthesis

Single crystals of $(Fe_{0.5}Co_{0.5})_{5-\delta}GeTe_2$ were grown by the chemical vapor transfer method[43,45], with iodine used as the transport agent. Elemental Fe (99.99%), Co (99.99%), and Te (99.999%) powders and small Ge pieces (99.999%), and were used in synthesis. The starting raw materials were mixed in the nominal molar ratio Fe:Co:Ge:Te = 3:3:1:2 inside a glovebox. The starting mixture was vacuumed and sealed inside a quartz tube, then reacted at 750 °C under isothermal conditions for 7 to 10 days. For MFM measurement, flakes were exfoliated and transferred to Si substrates using the tape method. For TEM measurement, flakes were transferred from Si substrates to SiN TEM windows with PDMS using standard 2D-material techniques.

### Magnetic force microscopy

Near-room-temperature MFM measurements were performed using an Asylum Research MFP-3D system. Low-temperature MFM

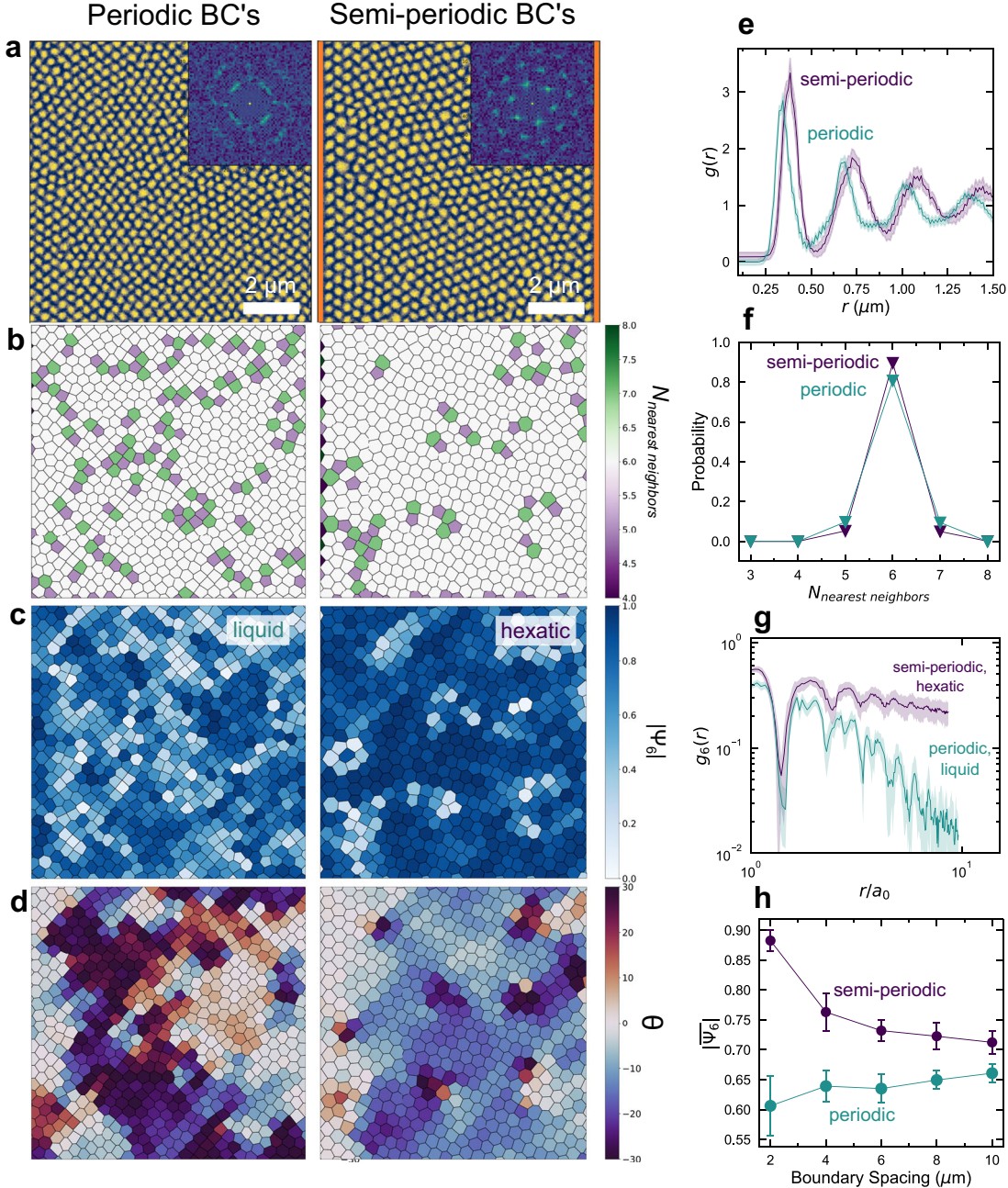

**Fig. 5 | Simulations of ordering in confined structures. a** Simulated real space images of the skyrmion lattices with periodic (left) and semi-periodic (right) boundary conditions. The fixed edges are shown in orange. Insets show the associated diffraction patterns. Scale bars are 2 μm. **b** Nearest neighbor map of the same images, where greater(less)-than-6 nearest neighbor sites are shown in green(-purple), and (**c**) bond orientational maps where the magnitude of $|\Psi_6(\mathbf{r})|$ is shown in blue. **d** Euler angle of the skyrmion lattice sites from −30° to +30°, with respect to the $x$ axis. Areas of similar color are mosaic domains which are rotationally uniform. **e** Radial distribution function, (**f**), histogram of the number of nearest neighbors, and (**g**), orientational correlation function $g_6(r)$, where the simulation with periodic boundary conditions shows liquid-like decay and the simulation with semi-periodic boundary conditions shows hexatic-like decay. Shaded regions correspond to standard deviation over 10 simulations. **h** Average orientational order as a function of aspect ratio and the space between the two aperiodic boundary conditions.

measurements were performed with a homemade variable-temperature MFM (Rutgers University). The tips were coated with nominally 100-nm Co by magnetron sputtering. The MFM signal (the shift of resonant frequency) is proportional to the out-of-plane stray field gradient, which was extracted by a phase-locked loop (SPECS). MFM images were taken with constant-height noncontact mode.

**Statistical analysis**

Statistical analysis was done in python using a combination of original code and the Freud[69] numerical statistics python package. Skyrmion locations were found using the scikit-image python package after

applying a Laplacian-of-Gaussian filter. Manual corrections were carried out where necessary. Delaunay triangulation and Voronoi tessellation were performed algorithmically in the scipy python package to define nearest neighbor skyrmions, $\mathbf{r}_{ij}$.

Pair distribution functions (PDF) were calculated as:

$$G(\mathbf{r}) = \frac{1}{N} \sum_{i,j} \delta\left(\mathbf{r} - \mathbf{r}_{ij}\right),$$

where $\mathbf{r}$ is a vector in real space, $N$ is the total number of skyrmions and $\mathbf{r}_{ij}$ is the vector between skyrmions $i$ and $j$. The structure factor was

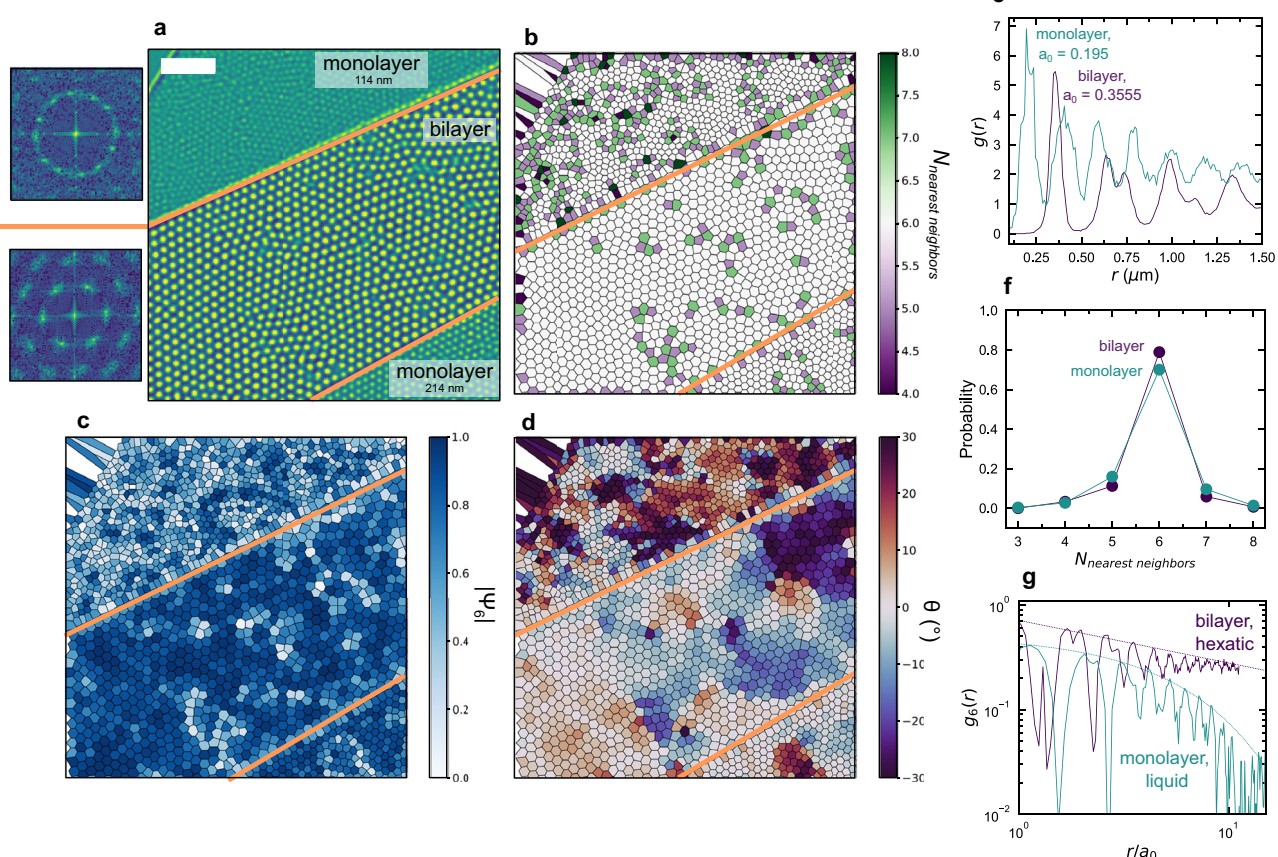

**Fig. 6 | Experimental observation of ordering. a** MFM micrograph of a sample with two, monolayer FCGT flakes and a rectangular, bilayer region of overlap. The edges of this region are shown in orange. The calculated structure factors are shown at left. Scale bar is 2 μm. **b** Nearest neighbor and (**c**) bond orientational maps of the same image. The monolayer regions can be characterized by long chains of topological defects, whereas the bilayer region forms an approximately single hexatic domain, commensurate with the direction of the confined edges. **d** Euler

angle, Θ, showing the rotational order of the skyrmion lattice sites. The monolayer region shows clear mosaic domains, while the confined, bilayer region is much more uniform. **e** Radial distribution function showing the increased lattice parameter of the bilayer region, indicating that it behaves semi-classically as a thicker flake of FCGT. **f** Histogram of the number of nearest neighbors. **g** $g_6(r)$, where the monolayer regions show a 2D liquid and the bilayer region shows hexatic behavior. Fits to exponential and algebraic decay are shown as dotted lines.

calculated from the PDF:

$$S_q(\mathbf{q}) = \sum_r e^{-iqr} G(\mathbf{r}),$$

where **q** is a vector in reciprocal space. Orientation independent radial distribution functions were calculated as:

$$g(r) = \frac{1}{2\pi r}\frac{1}{N}\sum_{i,j}\left\langle \delta\left(r - |\mathbf{r}_{ij}|\right)\right\rangle,$$

where $r$ is a distance. The calculations of $\Psi_6(\mathbf{r})$ and $g_6(r)$ are defined in the text.

## DF-(S)TEM

Dark-field (DF) transmission electron microscopy (TEM) experiments were carried out in a 2-beam condition, where the transmitted beam and a single diffracted beam allow for matching real space image features to defects with specific crystallographic orientations. Experimental data were collected on an FEI Titan ThemIS operating at 300 keV with a 0.68 m camera length and the diffraction patterns were indexed using a dynamical scattering simulation from the python Py4DSTEM package[70].

The four-dimensional (4D-) scanning transmission electron microscopy (STEM) experiments were performed using an electron

microscopy pixel array detector (EMPAD), where the 2D electron diffraction pattern was recorded over a 2D grid of real space probe positions, resulting in 4D datasets. Experimental data were acquired using a FEI Titan operated at 300 keV under field-free condition (Lorentz mode) with 150 pA beam current, 0.7 mrad semi-convergence angle, and a dwell time of 1 ms.

## Simulation

Numerical results were found using the micromagnetic simulation package Mumax3 running on the Lawrencium computing cluster at Lawrence Berkeley National Lab. The simulated area is a cuboid of dimensions 2500 nm × 2500 nm × 100 nm, divided into 5 nm × 5 nm × 5 nm cells. Effective medium material parameters used are $A = 5.5 \times 10^{-12}$ J m$^{-1}$, $D = 0.7$ mJ m$^{-2}$, $M_s = 301$ kA m$^{-1}$. The 5 nm cell size is below the exchange length given by $l_{ex} = \sqrt{\frac{2A}{\mu_\circ m_s^2}} \approx 9.8$ nm. To model the hexagonal order parameter as a function of different geometries, simulations were run with a range of aspect ratios from (500 nm × 2500 nm) to (2500 nm × 2500 nm) these simulations were performed with both fixed and semi-periodic boundary conditions (periodic on y boundary, fixed on x boundary). To quantify hexatic order in a statistically significant way a sample size of $N = 10$ different random seeds were used as initial conditions to position the nucleation sites of the Skyrmions. Simulations were relaxed from the initial state to reach a near-ground state condition and then advanced for 1000 ns to incorporate thermal ordering effects.

## Data availability

The MFM data generated in this study have been deposited in the Zenodo database under accession code 8011829. Other data used in these experiments are available from the authors upon reasonable request.

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

## Acknowledgements

P.M. and R.R. acknowledge funding from the Department of Defense, ARO Grant No. W911NF-21-2-0162 (ETHOS). This work was primarily funded by the U.S. Department of Energy, Office of Science, Office of Basic Energy Sciences, Materials Sciences and Engineering Division under Contract No. DE-AC02-05-CH11231 (Quantum Materials program KC2202). D.R. and P.F. are supported by the U.S. Department of Energy, Office of Science, Office of Basic Energy Sciences, Materials Sciences and Engineering Division under Contract No. DE-AC02-05-CH11231 (Nonequilibrium magnetic materials program MSMAG). R.Y. is supported by the National Science Foundation Graduate Research Fellowship under Grant No. DGE 2146752. This research used the Lawrencium computational cluster resource pro-vided by the IT Division at the Lawrence Berkeley National Laboratory, supported by the U.S. Department of Energy, Office of Science, Office of Basic Energy Sciences under Contract No. DE-AC02-05CH11231. The low-temperature MFM studies at Rutgers are supported by the Office of Basic Energy Sciences, Division of Materials Sciences and Engineering, U.S. Department of Energy under award no. DE-SC0018153. Y.T.S. and D.A.M. acknowledge funding support by the Department of Defense, Air Force Office of Scientific Research under award FA9550-18-1-0480. The elec-tron microscopy studies were performed at the Cornell Center for Mate-rials Research, a National Science Foundation (NSF) Materials Research Science and Engineering Center (DMR-1719875). The Cornell FEI Titan Themis 300 was acquired through NSF-MRI-1429155, with additional support from Cornell University, the Weill Institute and the Kavli Institute at Cornell. The authors thank M. Thomas, J. G. Grazul, M. Silvestry Ramos, K. Spoth for technical support and careful maintenance of the instruments.

## Author contributions

P.M., H.Z., R.R., and R.J.B. designed experiments. X.C., R.C., H.Z., and J.Y. synthesized samples. H.Z., Y.-T.C., and W.W. performed MFM measure-ments. R.Y., Y.T.S., and D.A.M. performed and analyzed (S)TEM imaging. D.R. implemented micromagnetic simulations. P.M. implemented sta-tistical analysis. P.M., H.Z., R.J.B., R.R., and P.F. analyzed results and wrote the paper. All authors have contributed to manuscript revisions.

## Competing interests

The authors declare no competing interests.
