## [Peer Review File · Nature Communications]

Reviewers' Comments:

Reviewer #1:

Remarks to the Author:

This manuscript reports magnetic force microscopy and 4D (S)TEM measurements on a bulk Co-doped FGT, which is ferromagnetic. FGT is a centrosymmetric ferromagnet, 50% Co alloying breaks spatial inversion symmetry, which in turn could lead to Dzyaloshinskii-Moriya interaction (DMI) sufficient to enable topological skyrmion type excitations of the ferromagnetic order. Microscopy was used to study the phase changes of skyrmions ensemble as a function of temperature and defects.

The manuscript reports engineering of an ordered skyrmion crystal through structural confinement on the micrometer scale, showing control over the order-disorder transition on scales relevant for device applications. The paper also discusses the behavior of 2D ensembles of magnetic skyrmions and the Hohenberg-Mermin-Wagner theorem, which states that true long-range, crystalline order cannot exist in 2D. The KTHNY theory is an extension of this and describes the melting of 2D objects with continuous symmetry. The paper shows the presence of crystallographic and/or magnetic disorder will interfere with the ideal KTHNY behavior, limiting ordering to short-range.

This is a very interesting topic, and it is a significant contribution to the field since the understanding and controlling ensembles of skyrmions are important for the realization of advanced computational devices based on skyrmion kinetics. Therefore I believe the manuscript should be considered for publication.

However, the reported study lacks considerably in thoroughness and consistency, as outlined below.

I therefore believe the authors should solve the following questions and concerns, which includes demonstration of repeatability of the results and to perform proper quantitative analysis of the various locations in a large area region. This is needed both to increase the technical quality of the manuscript and to ensure that the temperature dependence of topological defects is not unique to a chosen sample area.

List of questions and shortcomings:

1. For FCGT magnet epitaxy growth, is Co uniformly doped? Please help show the TEM image in a hundred micrometers range if possible.
2. Is the doping can be controlled during the chemical vapor transfer growth? That is, whether the formation of defects also shows a variation in different crystal locations? Since the images shown, for example in Fig. 2, at most are in the several tens of micrometer range. Also if the defects cannot be controlled during growth, the title should remove the word of "controlled".
3. Authors should show that the results are reproducible across different sample regions, address thickness dependence etc.

4. Fig 2g: where is the inset of lattice constant calculated from the first peak in f ?
5. For Fig. 6c, it seems the monolayer with 114 nm thickness and bilayer regions both have an approximately single hexatic domain. However, for Fig. 1b, it seems bilayer region has ordered skyrmions while monolayer shows disordered skyrmions. Please explain the discrepancy between these two images.
6. In line 11-12 page 5, the authors mentioned the skyrmions appears at 0 magnetic field. Is there any field training required? What is the reason for skyrmions at 0 magnetic field.
7. If authors actually believe this article will be read by a broader readership than people already working on skyrmions, they must define and explain terms like “polar magnet” and “4D-STEM”. Much of the text comes across as specialized, dense and technical.
8. How were the 5- or 7- fold sites determined. For the 5-sites shown in Fig. 2c, why it is not a 6-fold sites by connecting to the upper right point (in blue lines). Please comment on determination of 5-, 6- or 7 -fold sites.

9. Fig. 5 is missing a scale bar for the simulation in the real space.
10. Are the distorted skyrmions lattice supposed to be stable in time? Are they stable upon changing the scanning probe frequency or distance?
11. Why the thickness chosen to be 114 nm and 214 nm? Please explain.
12. There exist articles discussing skyrmions, bubbles and cross-over. Discuss how the presumed energy parameters/dopings in this system places it in the phase diagrams of skyrmions and bubbles.
13. Address whether authors think the skyrmion is thermodynamically stable or an excitation.

Reviewer #2:

Remarks to the Author:

In this manuscript, the authors have investigated Kosterlitz-Thouless-Halperin-Nelson-Young (KTHNY) behavior of skyrmion assembly and demonstrated the control of skyrmion assembly in a quasia-2D polar van-der Waals magnet $\text{Fe}_{0.5}\text{Co}_{0.5}\text{GeTe}_2$ (FCGT), in which the skyrmion formation have been recently reported by the authors' group [Sci. Adv. 8, eabm7103 (2022)]. KTHNY theory predicts nontrivial particle condensation in 2D systems and has been extensively studied in a variety of physical systems including colloids, liquid crystals and superconductor vortices. Recently, KTHNY behavior of skyrmion lattices has also been reported in several quasia-2D skyrmion materials [Nat. Nanotech 15, 761 (2020) and Adv. Fun. Mat. 30, 2004037 (2020)]. Overall, the experimental data, methodology, and analysis in this manuscript are sound. The manuscript can be divided into two parts. In the first half of the manuscript, the authors investigate KTHNY behavior in FCGT by analyzing the orientational correlation function $g_6(r)$, concluding the crystallographic defects make the skyrmion assembly a 2D liquid phase at all temperatures, which interpretation is also supported by EF-TEM images. The second half of the manuscript investigates the spatial confinement effect. On the basis of simulation and experimental results, the authors conclude that the hexatic phase of skyrmions is realized in the overlapping two thick flakes due to the spatial confinement effect. The conclusion of the first half of the manuscript is consistent with the result of Ref. 27 (claimed by the authors themselves in the main text), the novelty of this work lies in the second half of the manuscript (i.e., the investigation of confinement effect on KTHNY behavior of skyrmions). While the observation is interesting, it does not seem to be a significant phenomenon, and its generality and importance for physics or engineering is unclear. In addition, as itemized below, their discussion and interpretation of the confinement effect seem ambiguous and insufficient. Hence, it is difficult for me to recommend the publication in Nature Communications.

1) In the experiment for the confinement effect, "confinement sample" was fabricated by overlapping two flakes. I think this is unusual way to investigate the confinement effect of skyrmions. Why did the authors overlap two flakes to make a confinement sample instead of making a rectangle sample by using milling process? Does the observed phenomenon only occur in the overlapping sample of FCGT? Or can we observe a similar effect in other systems such as a rectangle single-layer flake or thin films that are fabricated by using milling process? These questions are also related to the significance of this work.

2) In Fig. 6, the authors mentioned that the monolayer region is the 2D liquid and the bilayer region is the hexatic phase. Consequently, it is implied that the number of defects in the monolayer region exceeds those in the bilayer region. Nevertheless, while the 214-nm monolayer (top) has many defects, the 114-nm monolayer (bottom) appears to have few defects (Fig. 3b). Moreover, the number of defects in the 144-nm monolayer is comparable to those in the bilayer region. Why does it seem difference in the number of defects between two monolayers exists? If dissimilarity is genuine, then the authors' conclusion that the confinement effect originates the hexatic phase becomes dubious.

3) The effects of sample boundaries (i.e. finite size effect) on KTHNY behavior has been a subject of discussion for a considerable duration discussed [for example, Rev. Mod. Phys. 60, 161 (1988), PRL 106, 235701 (2011), and J. Chem. Phys. 515 104702]. Usually, the finite size effect impedes the observation of true KTHNY behavior because KTHNY theory considers thermodynamic limit. Hence, it is nontrivial whether the skyrmion assembly observed in the confinement sample can be classified as the hexatic phase in the KTHNY theory or not, because the boundary effect heavily influences the condensation of skyrmions in the present case as the authors claimed. A more detailed discussion and verification of this point are necessary.

4) Minor point

-How much is the thickness of the sample in Fig. 2 and 3? As far as I read, there is no description of the thickness (I apologize if I overlooked it).

Reviewer #3:

Remarks to the Author:

The authors present a manuscript about ordered skyrmion phase achieved by a confined geometry in a polar vdW magnet. The topic is of current interest as several papers producing and observing ordered or semi-ordered skyrmion phases have been published recently. The authors themselves have been actively publishing in the field of skyrmions, exploring the FCGT system hosting skyrmions. The paper is well structured, referencing previous results and measuring further in a confined geometry, which results in a clearer evidence of a hexatic phase of the skyrmion lattice. The conclusions are well established and supported by the evaluation of experimental data, as well as theoretical simulations. However, the paper lacks details regarding the confinement itself and the physical underlying phenomena resulting in the observed hexatic phase. As the results pose an incremental addition to the understanding of the two-dimensional phase transition, I find it would be better suited in a different type of journal, e.g. Scientific Reports. Nevertheless, I would recommend the manuscript for publication after revision and response to following comments.

1. Please explain in detail in what sense does the confinement in the simulations result in a distinctively more ordered phase. The boundaries impose additional potential, but what form does it have? And is it long-range, does it influence the skyrmion in the centre of the confinement? More details on the confinement and its effect on the ordering would be desirable for strengthening the conclusions. If you assume an exponential potential of the boundary, could you change its amplitude to cause the high-ordered state. Could you get rid of the skyrmions near the boundary by forcing them more to the centre by engineering of the boundary itself. These are only several questions that arise from reading the manuscript and should be addresses in some way.
2. As you attribute the inability to achieve hexatic phase to kinetic pinning and the dislocations in the material, do you see a change in the strain while stacking the material to achieve the confinement conditions? Or how does that relate? And how does the changing of the temperature influence this?
3. After comment nr. 2, the question arises, how the potential is changed in the double-layer and the confinement case. In plain speech: What forces the skyrmions to be ordered better?
4. On page 10, last paragraph, continuing to page 11. I do not understand the statement of introducing the „new energy term“. This paragraph feels out of context for the presented work as it does not conclude of follow up this statement.
5. Page 14, Figure 6 e. The $g(r)$ peaks for the bilayer show distinct double peaks at the positions of the second and third maxima. The first peak of the monolayer shows similar behaviour. What does this mean for the ordering? Are there two distinct most probable positions of skyrmions being found in this distance? Is the double peak treated as second and third maxima? I understand that it does not change the decay characteristic of $g_6(r)$ presented in Figure 6g, but is still noteworthy.

REVIEWER COMMENTS

Responses are in blue

Reviewer #1 (Remarks to the Author):

This manuscript reports magnetic force microscopy and 4D (S)TEM measurements on a bulk Co-doped FGT, which is ferromagnetic. FGT is a centrosymmetric ferromagnet, 0% Co alloying breaks spatial inversion symmetry, which in turn could lead to Dzyaloshinskii-Moriya interaction (DMI) sufficient to enable topological skyrmion type excitations of the ferromagnetic order. Microscopy was used to study the phase changes of skyrmions ensemble as a function of temperature and defects.

The manuscript reports engineering of an ordered skyrmion crystal through structural confinement on the micrometer scale, showing control over the order/disorder transition on scales relevant for device applications. The paper also discusses the behavior of 2D ensembles of magnetic skyrmions and the Hohenberg-Mermin-Wagner theorem, which states that true long-range, crystalline order cannot exist in 2D. The KTHNY theory is an extension of this and describes the melting of 2D objects with continuous symmetry. The paper shows the presence of crystallographic and/or magnetic disorder will interfere with the ideal KTHNY behavior, limiting ordering to quasi-long-range.

This is a very interesting topic, and it is a significant contribution to the field since the understanding and controlling ensembles of skyrmions are important for the realization of advanced computational devices based on skyrmion kinetics. Therefore I believe the manuscript should be considered for publication. However, the reported study lacks considerably in thoroughness and consistency, as outlined below.

I therefore believe the authors should solve the following questions and concerns, which includes demonstration of repeatability of the results and to perform proper quantitative analysis of the various locations in a large area region. This is needed both to increase the technical quality of the manuscript and to ensure that the temperature dependence of topological defects is not unique to a chosen sample area.

We thank the reviewer for their favorable interpretation of our study and their helpful comments.

List of questions and shortcomings:

1. For FCGT magnet epitaxy growth, is Co uniformly doped? Please help show the TEM image in a hundred micrometers range if possible.

We thank the reviewer for the comment. Samples are transferred to Si substrates via the tape method from grown single crystals. We have added a statement in the manuscript in the introduction (P5L20) and relevant method section (P16L13) to make this more clear.

“For MFM measurement, flakes were exfoliated and transferred to Si substrates using the tape method. For TEM measurement, flakes were transferred from Si substrates to SiN TEM windows with PDMS using standard 2D-material techniques.”

The composition dependence of magnetization has been more thoroughly characterized in our previous works ^{1,2}, and we see no evidence of the segregation of Co. Measuring a large-field-of-view image with EDS also does not show any evidence of Co segregation:

Figure R1 | large-field-of-view micrograph with EDS mapping, showing no evidence of chemical segregation on the scale of the skyrmions

2. Is the doping can be controlled during the chemical vapor transfer growth? That is, whether the formation of defects also shows a variation in different crystal locations? Since the images shown, for example in Fig. 2, at most are in the several tens of micrometers range. Also if the defects cannot be controlled during growth, the title should remove the word of “controlled”.

The reviewer raises a good question here. We have minimal control over the chemical defects during the chemical vapor transfer growth, as during growth, we need to quench the sample to stabilize the polar structure. This will kinetically freeze chemical and crystallographic defects into the crystal, directly leading to phenomena like the strain fields shown in Figure 4. Thus, the presence of these structural defects conflicts with our engineering of ordering in the skyrmion lattice. This is our primary argument to why we

are not able to observe an ordered phase in the samples without finite size effects. The term “controlled” in the title is in reference to the ability to force ordering of the skyrmions through mechanical confinement.

3. Authors should show that the results are reproducible across different sample regions, address thickness dependence etc.

We thank the review for the comment. The observed trends hold over different locations and samples, showing liquid-like behavior when the areas are unconfined such as in monolayer flakes and bilayer flakes only bounded on one side, and ordered hexatic-like behavior when bilayer regions are bounded on both sides. In cases where the population size is small, or the domain is strangely shaped, the statistical analysis starts to break down and become less meaningful, which is why these data were not presented in the manuscript. The following figure has been added to the SI (**Sup. Figure 7**) to further support our assertion and make this more clear.

Figure R2 | MFM images, Voronoi maps, Ψ_6 maps, structure factors (SF) and pair distribution functions (PDF) of several other monolayer flake samples and locations, showing that the observed trends hold. In all cases, the monolayer flakes show liquid-like behavior and the $|\Psi_6|$ is in a regime below that of the confined bilayer. Samples 1 and 2 are 143 nm thick, samples 3 and 4 are 185 nm thick, and sample 5 is 207 nm thick.

4. Fig 2g: where is the inset of lattice constant calculated from the first peak in f ?
 We thank the reviewer for noticing- this part of the caption was left from a previous version of the figure, but the inset was removed. The caption has been fixed to match the figure.

5. For Fig. 6c, it seems the monolayer with 114 nm thickness and bilayer regions both have an approximately single hexatic domain. However, for Fig. 1b, it seems bilayer

region has ordered skyrmions while monolayer shows disordered skyrmions. Please explain the discrepancy between these two images.

Figures 1b and 6a are the same image, just rotated 180 degrees to facilitate the illustration in 1c. **Figure 1** has been changed to avoid this confusion.

Figure 1 | Layered polar magnet. **a** The unit cell of FCGT, showing the AA' stacking which breaks inversion symmetry and allows for a nonzero DMI. **b** Sample with Néel skyrmions showing multiple regimes of, **c**, ordered (close packed) and disordered skyrmion configurations. The scale bar is 2 μm .

In this image, in both monolayer regions, the skyrmions can be observed to be in a disordered phase through N_{nn} and $|\Psi_6|$, but this was not explicitly discussed in the manuscript because the small number of skyrmions in the image makes the calculation not statistically significant.

Figure R3 | MFM image showing monolayer and bilayer FCJT regions with the corresponding histogram for N_{nn} and $|\Psi_6|$ values. These values indicate a disordered phase in the lower-right 214 nm monolayer region, which is consistent with the explicitly disordered top left 114 nm region, but the statistics cannot be rigorously defined because of the small population size.

Figure R3 has been added to the SI (**Sup. Figure 11**).

We can also extend this effect to other measured bilayer regions, as shown above in our response to question 3 (**Figure R2**)

6. In line 11-12 page 5, the authors mentioned the skyrmions appears at 0 magnetic field. Is there any field training required? What is the physical origin for emergence of skyrmions at 0 magnetic field.

We thank the reviewer for the comment. There are two reasons for the stabilization of the skyrmion lattice at zero magnetic field. From the structure, the strain in the flake, which is confirmed by the TEM, can help stabilize the skyrmion lattice at zero magnetic field³. On the other hand, the stray field of the PFM tips can induce the formation of metastable skyrmion lattice⁴. In the FCJT system, we don't need to special train for most of the samples, probably because a small field is needed to stabilize the skyrmion lattice. The skyrmion lattice can be formed directly during the first scanning in the thin nanoflakes (110~300 nm).

7. If authors actually believe this article will be read by a broader readership than people already working on skyrmions, they must define and explain terms like "polar magnet" and "4D-STEM". Much of the text comes across as specialized, dense and technical.

We thank the reviewer for the comment. We define a polar magnetic metal as “a metallic magnet which intrinsically breaks spatial inversion symmetry.” This has been added to the text (P5L9).

In the method section, we describe 4DSTEM as “The four-dimensional (4D-) scanning transmission electron microscopy (STEM) experiments were performed using an electron microscopy pixel array detector (EMPAD), where the 2D electron diffraction pattern was recorded over a 2D grid of real space probe positions, resulting in 4D datasets.” We have added a “see methods” note in the relevant place in the text (P10L16).

8. How were the 5- or 7- fold sites determined. For the 5-sites shown in Fig. 2c, why it is not a 6-fold sties by connecting to the upper right skyrmion. Please comment on determination of 5-, 6- or 7 -fold sites.

We thank the reviewer for the comment. The number of nearest neighbors was calculated from the centroids algorithmically in python using Delaunay triangulation and Voronoi decomposition. Delaunay triangulation finds triangles of 3 points such that no other point is within the circumference of the circle defined by these points. This network of “bonds” then corresponds to the dual graph of the Voronoi map, which defines cells around points where the edges are exactly halfway between adjacent points. As part of this calculation, N_{nn} is rigorously defined. This calculation is efficiently implemented in scipy, which is a statistical standard and was used here. A sentence has been added to the relevant method section (P16L31) to be more descriptive.

“Delaunay triangulation and Voronoi tessellation were performed algorithmically in the scipy python package to define nearest neighbor skyrmions, \mathbf{r}_{ij} .”

Figure R4 | Example Delaunay triangulation and Voronoi plot of the monolayer FCGT sample measured at 306 K in order to demonstrate the calculation of N_{nn} .

9. Fig. 5 is missing a scale bar for the simulation in the real space.

We thank the review for the comment. In **Figure 5**, the cell sides are $10\mu\text{m}$ which was defined in the caption- a scale bar has been added instead for consistency with other figures.

Figure 5 | Simulations of ordering in confined structures. **a** Simulated real space images of the skyrmion lattices with periodic (left) and semi-periodic (right) boundary conditions. The fixed edges are shown in orange. Insets show the associated diffraction patterns. Scale bars are $2\mu\text{m}$. **b** Nearest neighbor map of the same images, where greater(less)-than-6 nearest neighbor sites are shown in green(purple), and **c** bond orientational maps where the magnitude of $|\Psi_6(r)|$ is shown in blue. **d** Euler angle of the skyrmion lattice sites from -30° to $+30^\circ$, with respect to the x axis. Areas of similar color are mosaic domains which are rotationally uniform. **e** Radial distribution function, **f**, histogram of the number of nearest neighbors,

and **g**, orientational correlation function $g_6(r)$, where the simulation with periodic boundary conditions shows liquid-like decay and the simulation with semi-periodic boundary conditions shows hexatic-like decay. Shaded regions correspond to standard deviation over 10 simulations. **h**, Average orientational order as a function of aspect ratio and the space between the two aperiodic boundary conditions.

10. Are the distorted skyrmions lattice supposed to be stable in time? Are they stable upon changing the scanning probe frequency or distance?

Because the acquisition time of the MFM images is so long, we can say that the skyrmion configurations are explicitly stable for hours to days. Because the skyrmion lattice should be very sensitive to environmental changes, experiments longer than this are very impractical, since the sample would have to stay in the microscope in a completely controlled environment.

Regarding the skyrmions themselves, we have observed previously “set” skyrmions after as long as 3 months via MFM, showing that the skyrmions themselves are stable for long periods of time.

Figure R5 | **a** MFM scan of an FCNT bilayer, showing the stability of skyrmions after ~3 months. **b** MFM scan of a monolayer flake with two different scan speeds (0.4 Hz and 0.3 Hz), showing the stability of the ensemble under scan rate. Scale bars are 2 μm in both panels.

11. Why the thickness chosen to be 114 nm and 214 nm? Please explain.

In FCNT, the skyrmions only exist in monolayers in a range of thicknesses from ~110-300 nm. Because the samples are exfoliated using the tape method and the unit cell of FCNT is large, incredibly precise control of thickness is difficult, but control within this range is more achievable.

Figure 2 from ⁵, showing stable skyrmions between ~ 125 - 250 nm thicknesses.

In the bilayer shown in **Figure R5**, the flakes are $\sim 1.6 \mu\text{m}$ and 135 nm thick. As this thicker flake is far out of the range where we see the skyrmions, we cannot obtain the skyrmion lattice but a superposition of both the stripe domains and the skyrmion structure in the thinner flake.

Figure R6 | Bilayer built of $1.6 \mu\text{m}$ and 135 nm FCGT flakes, where the thicker flake is far outside the range of stable skyrmions and so we cannot resolve the same skyrmion lattice as in Figure 6.

12. There exist articles discussing skyrmions, bubbles and crossover. Discuss how the

presumed energy parameters/dopings in this system places it in the phase diagrams of skyrmions and bubbles.

In previous papers, in pure Fe_5GeTe_2 , one can observe the skyrmions, bubbles even crossover⁶⁻⁸. The pure Fe_5GeTe_2 of structure is complex and sensitive to the thermal history and growth process. In principle, the Fe_5GeTe_2 has a centrosymmetric structure with ABC stacking. However, the Fe_5GeTe_2 hosts a local breaking bulk or interface inversion symmetry due to the Fe1 site ordering⁹ and the emergence of the AA phase after the thermal cycles¹⁰, respectively. Thus, due to the competition of the weak local DMI interaction, magnetic anisotropy energy, and dipolar energy, the pure Fe_5GeTe_2 shows a complex chiral domain wall. In our case, due to the strong DMI interaction induced by breaking bulk inversion symmetry, FCGT shows the skyrmion lattice at ambient conditions.

We have also previously carried out thickness dependent LTEM measurements on FCGT which, in all observed cases, the domain walls are Néel type, and the Néel type skyrmions can be stabilized in the ~100-200 nm nanoflake under a finite magnetic field. As the temperature decreases, the perpendicular anisotropy and DMI will increase¹¹. LTEM measurements at low temperatures show that the character of the domain wall at low temperatures are still Néel-type.

13. Address whether authors think the skyrmion is thermodynamically stable or an excitation.

As mentioned above in our answer to question 10, the skyrmions themselves are stable at ambient conditions for long periods of time (>3 months). The skyrmion lattice we believe to be kinetically limited by sample defects, limiting condensation of the ordered phase.

Reviewer #2 (Remarks to the Author):

In this manuscript, the authors have investigated Kosterlitz-Thouless-Halperin-Nelson-Young (KTHNY) behavior of skyrmion assembly and demonstrated the control of skyrmion assembly in a quasia-2D polar van-der Waals magnet $\text{Fe}_{0.5}\text{Co}_{0.5}\text{GeTe}_2$ (FCGT), in which the skyrmion formation have been recently reported by the authors' group [Sci. Adv. 8, eabm7103 (2022)]. KTHNY theory predicts nontrivial particle condensation in 2D systems and has been extensively studied in a variety of physical systems including colloids, liquid crystals and superconductor vortices. Recently, KTHNY behavior of skyrmion lattices has also been reported in several quasia-2D skyrmion materials [Nat. Nanotech 15, 761 (2020) and Adv. Fun. Mat. 30, 2004037 (2020)].

Overall, the experimental data, methodology, and analysis in this manuscript are sound. The manuscript can be divided into two parts. In the first half of the manuscript, the authors investigate KTHNY behavior in FCGT by analyzing the orientational correlation function $g_6(r)$, concluding the crystallographic defects make the skyrmion assembly a 2D liquid phase at all temperatures, which interpretation is also supported by EF-TEM images.

The second half of the manuscript investigates the spatial confinement effect. On the basis of simulation and experimental results, the authors conclude that the hexatic phase of skyrmions is realized in the overlapping two thick flakes due to the spatial confinement effect.

The conclusion of the first half of the manuscript is consistent with the result of Ref. 27 (claimed by the authors themselves in the main text), the novelty of this work lies in the second half of the manuscript (i.e., the investigation of confinement effect on KTHNY behavior of skyrmions). While the observation is interesting, it does not seem to be a significant phenomenon, and its generality and importance for physics or engineering is unclear. In addition, as itemized below, their discussion and interpretation of the confinement effect seem ambiguous and insufficient. Hence, it is difficult for me to recommend the publication in Nature Communications.

We thank the reviewer for their helpful comments on our manuscript.

1) In the experiment for the confinement effect, “confinement sample” was fabricated by overlapping two flakes. I think this is unusual way to investigate the confinement effect of skyrmions. Why did the authors overlap two flakes to make a confinement sample instead of making a rectangle sample by using milling process? Does the observed phenomenon only occur in the overlapping sample of FCGT? Or can we observe a similar effect in other systems such as a rectangle single-layer flake or thin films that are fabricated by using milling process? These questions are also related to the significance of this work.

We agree with the reviewer that layering flakes in this way to create spatial confinement is nonstandard, but we pursued this avenue because the fragility of the material makes traditional lithography processes difficult, and working to solve this challenge is a very active area in 2d materials research^{12–15}. In principle, we believe that a lithographically confined sample could show similar behavior, and we tried exactly this experiment, but in our samples ion milling of the FCGT flakes introduces defects in the sample from heating and knock-on damage that drastically affects magnetic properties.

Figure R7 | AFM images illustrating milling damage at the edge of the FCGT flakes

Additionally, even the additional director field coming from the spatial confinement may not be enough to actually order the skyrmions in monolayer samples. KTHNY behavior is disrupted by the nonuniform strain field within the flake and in our manuscript, we conclude that this is the reason we never observe the ordered phases of the skyrmion lattice. The reason that this spatial confinement may still not work in monolayer samples, but does in the bilayer samples, is that this strain in the bilayer is averaged across the two flakes, potentially reducing the composite nonuniformity. This composite averaging of properties can also be observed with the equilibrium lattice constant of the skyrmions in the bilayer, where we see the anisotropies of the individual layers average out in the bilayer system and behave like a single sample.

To the end of reproducibility, we have observed this ordering phenomena in other bilayer flakes, where skyrmions are ordered against the edge of the overlapping region, nucleating or stabilizing the ordered phase. We have added the following figure (**Figure R8**) to the SI (**Sup. Figure 12**) to show this more clearly.

The following text has also been added to the manuscript (P15 L19):

“Another factor to this difference may be that strain and pinning defects in the bilayer are averaged across the two flakes, potentially reducing the composite nonuniformity. This averaging of properties can also be observed with the equilibrium lattice constant of the skyrmions in the bilayer, where we see the anisotropies of the individual layers average out in the bilayer system and behave like a single sample.”

Figure R8 | MFM images, Voronoi maps, Ψ_6 maps, structure factors (SF) and pair distribution functions (PDF) of other bilayer samples and locations, showing that the observed trends hold. Samples show liquid-like behavior when the areas are unconfined, such as in monolayer flakes and bilayer flakes only bounded on one side, and ordered hexatic-like behavior when bilayer regions are bounded on both sides. Qualitatively, we observe that skyrmions always order along the edge of the flake, supporting our assertion that the ordering field from the confinement is trapping or nucleating skyrmions, which increase the stability of nearby structural domains.

2) In Fig. 6, the authors mentioned that the monolayer region is the 2D liquid and the bilayer region is the hexatic phase. Consequently, it is implied that the number of defects in the monolayer region exceeds those in the bilayer region. Nevertheless, while the 214-nm monolayer (top) has many defects, the 114-nm monolayer (bottom) appears to have few defects (Fig. 3b). Moreover, the number of defects in the 144-nm monolayer is comparable to those in the bilayer region. Why does it seem difference in the number of defects between two monolayers exists? If dissimilarity is genuine, then the authors' conclusion that the confinement effect originates the hexatic phase becomes dubious.

We thank the reviewer for the comment. Because the lower-right region in the image is such as small population (~ 180 skyrmions), it is difficult to say anything statistically meaningful. From what we do feel confident saying (counting the number of defects, $|\Psi_6|$),

we do see that the lower-right region follows the trend dictated by the upper-left region. In the image shown in Figure 6, in both monolayer regions, the skyrmions can be observed to be less ordered through N_{nn} and $|\Psi_6|$, but this was not explicitly discussed in the manuscript because the small number of skyrmions in the bottom right region makes this calculation not statistically significant. Physically, we would expect a similar mechanism to the bilayer to happen in a monolayer near an edge, where skyrmions stick to the edge, but the skyrmions are much smaller, so the coherence length is also much smaller. e.g. to reach the ordered phase, we would have to have a confined flake with straight edges on the order of ~ 25 skyrmion radii.

Figure R3 | MFM image showing monolayer and bilayer FCGT regions with the corresponding histogram for N_{nn} and $|\Psi_6|$ values. These values indicate a disordered phase in the lower-right 214 nm monolayer region, which is consistent with the explicitly disordered top left 114 nm region, but the statistics cannot be rigorously defined because of the small population size.

We agree that the phenomenon could be somewhat ambiguous derived from only one image and we have included more monolayer and bilayer examples. As noted above, this correlates with the skyrmions “sticking” to the edge of the domain, either nucleating or getting trapped there, and allowing the stabilization of a more ordered phase.

Figure R2 | MFM images, Voronoi maps, Ψ_6 maps, structure factors (SF) and pair distribution functions (PDF) of several other monolayer flake samples and locations, showing that the observed trends hold. In all cases, the monolayer flakes show liquid-like behavior and the $|\Psi_6|$ is in a regime below that of the confined bilayer. Samples 1 and 2 are 143 nm thick, samples 3 and 4 are 185 nm thick, and sample 5 is 207 nm thick.

The observed trends hold over different locations and samples, showing liquid-like behavior when the areas are unconfined such as in monolayer flakes and bilayer flakes only bounded on one side, and ordered hexatic-like behavior when bilayer regions are bounded on both sides. In cases where the population size is small, or the domain is strangely shaped, the statistical analysis starts to become less meaningful, which is why these data were not presented in the manuscript. **Figures R1 and R5** (monolayers and bilayers) have been added to the SI (**Sup. Figures 7 and 12**) for clarity.

3) The effects of sample boundaries (i.e. finite size effect) on KTHNY behavior has been a subject of discussion for a considerable duration discussed [for example, Rev. Mod. Phys. 60, 161 (1988), PRL 106, 235701 (2011), and J. Chem. Phys. 515 104702]. Usually, the finite size effect impedes the observation of true KTHNY behavior because KTHNY theory considers thermodynamic limit. Hence, it is nontrivial whether the skyrmion assembly observed in the confinement sample can be classified as the hexatic phase in the KTHNY theory or not, because the boundary effect heavily influences the condensation of skyrmions in the present case as the authors claimed. A more detailed discussion and verification of this point are necessary.

We absolutely agree and thank you for pointing this out. KT theory is true in the thermodynamic (infinite) limit, thus the confined region is explicitly not pure-KTHNY behavior. Importantly noted by ¹⁶⁻¹⁸, as well as ¹⁹⁻²¹, the liquid-hexatic-phase transition can still be observed in systems where nonlinear potentials and finite size effects are present. In our case, it is not our intent to downplay the importance of these finite size effects- in fact the opposite, where we believe these size effects are responsible for the stabilization of the ostensibly ordered phase, overcoming the kinetic freezing of the intrinsic defects. Here, where the KTHNY framework is more directly considered in the first part of the paper, it is explicitly broken by our spatial confinement but still provides a useful framework for the interpretation of the order-disorder transition.

We have reworked part of the text to make this statement more explicit (P13L25):

“We note, however, that in our engineered systems KTHNY theory is no longer directly applicable. KTHNY theory is true in the infinite limit, thus the confined region is explicitly not pure-KTHNY behavior. Notably, however, the liquid-hexatic-phase transition can still be observed in systems where nonlinear potentials and finite size effects are present^{17,18,20,21}, making the KTHNY terminology a useful framework for the interpretation of the order-disorder transition.”

4) Minor point-How much is the thickness of the sample in Fig. 2 and 3? As far as I read, there is no description of the thickness (I apologize if I overlooked it).

Thank you for pointing out the oversight- the flake is ~125 nm thick. This has been noted in the manuscript (P7L1).

Reviewer #3 (Remarks to the Author):

The authors present a manuscript about ordered skyrmion phase achieved by a confined geometry in a polar vdW magnet. The topic is of current interest as several papers producing and observing ordered or semi-ordered skyrmion phases have been published recently. The authors themselves have been actively publishing in the field of skyrmions, exploring the FCGT system hosting skyrmions. The paper is well structured, referencing previous results and measuring further in a confined geometry, which results in a clearer evidence of a hexatic phase of the skyrmion lattice. The conclusions are well established and supported by the evaluation of experimental data, as well as theoretical simulations. However, the paper lacks details regarding the confinement itself and the physical underlying phenomena resulting in the observed hexatic phase. As the results pose an incremental addition to the understanding of the two-dimensional phase transition, I find it would be better suited in a different type of journal, e.g. Scientific Reports. Nevertheless, I would recommend the manuscript for publication after revision and response to following comments.

We thank the reviewer for their helpful comments on our manuscript

1. Please explain in detail in what sense does the confinement in the simulations result in a distinctively more ordered phase. The boundaries impose additional potential, but what form does it have? And is it long-range, does it influence the skyrmion in the centre of the confinement? More details on the confinement and its effect on the ordering would be desirable for strengthening the conclusions. If you assume an exponential potential of the boundary, could you change its amplitude to cause the high-ordered state. Could you get rid of the skyrmions near the boundary by forcing them more to the centre by engineering of the boundary itself. These are only several questions that arise from reading the manuscript and should be addresses in some way.

We appreciate the reviewer asking this question. We were thinking the same, about the local ordering as a function of distance from the boundary. This was not included in the manuscript, however, because our results indicate that the population of skyrmions is not large enough to say anything meaningful (**Figure R9d,e**).

Concerning the height of the potential at the boundary, our simulations consider the end members, namely an infinite or 0 potential at $x=(0,1)$, and if those are not ordered, we have no reason to expect a finite potential will be. Making these infinite potentials closer does increase the ordering of the system, which we show in **Figure 5** and **Sup. Figure 8**, but when the population is decreased, at a certain point the statistics become not meaningful. This is also the reason for **Figure 5h**, which implicitly shows that the proximity to the boundary conditions increases ordering in a nonlinear way.

Because of the way the simulation is built, having a finite potential inside the cell is experimentally ambiguous- we can accurately simulate having no boundary or having no material as 0 and infinite potentials respectively, but ascribing a value to the potential no longer necessarily corresponds to the real system. It is possible to implement a finite anisotropy rather than a fixed $m=(0,0,1)$ condition (infinite potential). To demonstrate, we adjust the boundary anisotropy between 0 mJ/m^3 and $1\text{e}6 \text{ mJ/m}^3$ by steps of 0.1mJ/m^3 . For the low anisotropy finite potential condition, domain walls and skyrmions become attached to the boundary in a way which is not reflected in the data, showing that an infinite or high potential is a good model for our system.

The periodic boundary condition treats the nearest neighbor exchange terms (Heisenberg, DMI) by coupling cells on opposite sides of the simulated volume in the same way as neighboring spins in the bulk. The long-range dipolar interaction is accommodated in the periodic boundary conditions by using the principle of superposition to calculate an appropriate image charge for the simulated area. In the simulation we specify an input of one image charge, meaning that the simulation “sees” an additional dipolar field equivalent to a neighboring simulated volume equal to the size of the actual simulated volume.

Figure R9 | Plots showing $|\Psi_6|$, **a**, and P_6 , **b**, as a function of finite boundary anisotropy between 0 and $1\text{e}6 \text{ mJ m}^{-3}$. Out-of-plane anisotropy is introduced at the left and right boundaries of the cell. **c**, plot showing $|\Psi_6|$ in the bilayer sample, the same as Figure 6c in the main text, rotated to showcase the calculation in d,e. $|\Psi_6|$ can then be binned and plotted as a function of x coordinate, showcasing distance from the boundary, in both simulated (**d**) and measured (**e**) cells.

2. As you attribute the inability to achieve hexatic phase to kinetic pinning and the dislocations in the material, do you see a change in the strain while stacking the material

to achieve the confinement conditions? Or how does that relate? And how does the changing of the temperature influence this?

While an interesting question, this is unfortunately not an achievable experiment. Experimental measures of strain come from 4D STEM, which we have seen is not achievable on the bilayer system.

In the bilayer FCGT, the flakes are not in atomic contact with one another- in cross section, we observe ~5 nm of organic debris between the flakes, which is a large reason that we claim any coupling should be magnetostatic, instead of electrical or elastic. In this case, it would be surprising if the crystallographic defects in one layer have any influence on the other. This is also discussed in **Sup Figure 10**, where we can simulate a bilayer with only magnetostatic coupling to see that the skyrmions track across this 5 nm gap.

The following text has also been added to the manuscript (P15 L19):

“Another factor to this difference may be that strain and pinning defects in the bilayer are averaged across the two flakes, potentially reducing the composite nonuniformity. This averaging of properties can also be observed with the equilibrium lattice constant of the skyrmions in the bilayer, where we see the anisotropies of the individual layers average out in the bilayer system and behave like a single sample.”

Figure R910 | a HAADF cross section of an FCGT bilayer showing ~ 5 nm of organic debris between the two. Because of this, we expect any coupling between the two to be magnetostatic and not elastic or electrical. **b (Sup. Figure 9)** Simulation showing stable skyrmions in the bilayer system with a 5 nm gap between the layers, where skyrmions track from one layer to the other, indicating that the bilayer behaves as a single magnet.

3. After comment nr. 2, the question arises, how the potential is changed in the double-layer and the confinement case. In plain speech: What forces the skyrmions to be ordered better?

Ordering of the director field in liquid crystals through spatial confinement is a well-established area of research and was one of the first principles ever established in the field^{22–26}. In these materials, it is argued that the ordering field is elastic/steric, as the large size of the molecules facilitates packing along edges. In the last few years, this idea of spatial confinement has been touched upon in magnetic skyrmions²⁷, with studies considering traditional lithography techniques^{28,29}, intrinsically occurring grain boundaries³⁰, and elastic strain³¹. In magnetic skyrmions, interactions with the sample edge are likely due to magnetoelastic or shape anisotropies, which we suspect is also the case in our system. In the steady state, we would expect an ordering of the skyrmions at the edge to minimize stray magnetic field/interaction with the large potential at the boundary (e.g. shape anisotropy), favoring a locally ordered configuration. If this interaction energy is a primary driving force for ordering, we would expect the local order parameter to decay with whatever term governs the interaction at the boundary, for instance $1/d^2$, if the interaction were magnetostatic. From our simulations (**Figure 5h** and **Sup Figure 8**), we estimate this decay length as 10-15 lattice constants. As discussed above, however, we are not able to observe this decay of the ordering constant, potentially due to the small population of skyrmions.

We have added the following text to the SI (**Sup Figure 8**):

“In magnetic skyrmions, interactions with the sample edge are likely due to magnetoelastic or shape anisotropies, which we suspect is also the case in our system. In the steady state, we would expect an ordering of the skyrmions at the edge to minimize stray magnetic field/interaction with the large potential at the boundary (e.g. shape anisotropy), favoring a locally ordered configuration.”

4. On page 10, last paragraph, continuing to page 11. I do not understand the statement of introducing the „new energy term“. This paragraph feels out of context for the presented work as it does not conclude or follow up this statement.

Thank you for pointing this out. This was intended to be a transition to the discussion about the director field imposed by the boundary conditions, but we have reworked the paragraph to be more precise.

Our conclusion is also in agreement with the results in ref. ³², in which the authors also do not observe a true hexatic phase in a magnetic skyrmion lattice, indicated by exponential decay in $g_6(r)$, but attribute it to the arbitrarily long timescales associated with condensation. If this is true and condensation of the hexatic and solid phases is not achievable on a reasonable time scale due to pinning, can we engineer the introduction of a new field that favors ordering to overcome these kinetic limitations and influence formation of the ordered phase? Intuitively, this should be possible, as it is analogous to other disordered functional systems such as relaxor ferroelectrics³³ and spin glasses³⁴ where strong fields can force an otherwise frozen/disordered order parameter into a long-range ordered state³⁵.

5. Page 14, Figure 6 e. The $g(r)$ peaks for the bilayer show distinct double peaks at the positions of the second and third maxima. The first peak of the monolayer shows similar behaviour. What does this mean for the ordering? Are there two distinct most probable positions of skyrmions being found in this distance? Is the double peak treated as second and third maxima? I understand that it does not change the decay characteristic of $g_6(r)$ presented in Figure 6g, but is still noteworthy.

What look like bifurcated peaks are two peaks each and come from the 2nd/ 3rd and 4th/5th nearest neighbors in a hexagonal configuration. The width of these peaks is related to the precision of the local lattice constants in the skyrmion crystal.

Figure R11 | **a** Illustration of the nearest neighbors in a hexagonal packing. **b** $g_6(r)$ from Figure 6g, plotted on a linear scale in r to more clearly show the positions of the nn peaks. The first 8 nn's shown in **a** are marked.

References

1. Chen, X. *et al.* Pervasive beyond Room-Temperature Ferromagnetism in a Doped van der Waals Magnet. *Phys. Rev. Lett.* **128**, 217203 (2022).
2. Zhang, H. *et al.* A room temperature polar magnetic metal. *Phys. Rev. Materials* **6**, 044403 (2022).
3. Feng, C. *et al.* Field-Free Manipulation of Skyrmion Creation and Annihilation by Tunable Strain Engineering. *Advanced Functional Materials* **31**, 2008715 (2021).
4. Zhang, S. *et al.* Direct writing of room temperature and zero field skyrmion lattices by a scanning local magnetic field. *Appl. Phys. Lett.* **112**, 132405 (2018).
5. Zhang, H. *et al.* Room-temperature skyrmion lattice in a layered magnet (Fe_{0.5}Co_{0.5})₅GeTe₂. *Science Advances* **8**, eabm7103 (2022).
6. Zhang, C. *et al.* Magnetic Skyrmions with Unconventional Helicity Polarization in a Van Der Waals Ferromagnet. *Advanced Materials* **34**, 2204163 (2022).
7. Schmitt, M. *et al.* Skyrmionic spin structures in layered Fe₅GeTe₂ up to room temperature. *Commun Phys* **5**, 1–9 (2022).
8. Gao, Y. *et al.* Manipulation of topological spin configuration via tailoring thickness in van der Waals ferromagnetic Fe₅GeTe₂. *Phys. Rev. B* **105**, 014426 (2022).
9. Ly, T. T. *et al.* Direct Observation of Fe-Ge Ordering in Fe_{5-x}GeTe₂ Crystals and Resultant Helimagnetism. *Advanced Functional Materials* **31**, 2009758 (2021).
10. Chen, X. *et al.* Thermal cycling induced alteration of the stacking order and spin-flip in the room temperature van der Waals magnet Fe₅GeTe₂. Preprint at <https://doi.org/10.48550/arXiv.2209.04560> (2022).
11. Schlotter, S., Agrawal, P. & Beach, G. S. D. Temperature dependence of the Dzyaloshinskii-Moriya interaction in Pt/Co/Cu thin film heterostructures. *Appl. Phys. Lett.* **113**, 092402 (2018).
12. Andersen, A. B., Shivayogimath, A., Booth, T., Kadkhodazadeh, S. & Hansen, T. W. Limiting Damage to 2D Materials during Focused Ion Beam Processing. *physica status solidi (b)* **257**, 2000318 (2020).
13. Cheng, Z. *et al.* Convergent ion beam alteration of 2D materials and metal-2D interfaces. *2D Mater.* **6**, 034005 (2019).
14. Sarcan, F. *et al.* Understanding the impact of heavy ions and tailoring the optical properties of large-area Monolayer WS₂ using Focused Ion Beam. Preprint at <https://doi.org/10.48550/arXiv.2210.04315> (2022).
15. Barcons Ruiz, D. *et al.* Engineering high quality graphene superlattices via ion milled ultra-thin etching masks. *Nat Commun* **13**, 6926 (2022).
16. Strandburg, K. J. Two-dimensional melting. *Rev. Mod. Phys.* **60**, 161–207 (1988).
17. Kapral, R. Progress in the Theory of Mixed Quantum-Classical Dynamics. *Annual Review of Physical Chemistry* **57**, 129–157 (2006).
18. Prestipino, S., Saija, F. & Giaquinta, P. V. Hexatic Phase in the Two-Dimensional Gaussian-Core Model. *Phys. Rev. Lett.* **106**, 235701 (2011).
19. Fisher, D. S., Halperin, B. I. & Morf, R. Defects in the two-dimensional electron solid and implications for melting. *Phys. Rev. B* **20**, 4692–4712 (1979).
20. Udink, C. & van der Elsken, J. Determination of the algebraic exponents near the melting transition of a two-dimensional Lennard-Jones system. *Phys. Rev. B* **35**, 279–283 (1987).
21. Joos, B. & Duesbery, M. S. Dislocation energies in rare-gas monolayers on graphite. *Phys Rev Lett* **55**, 1997–2000 (1985).

22. Tortora, L. & Lavrentovich, O. D. Chiral symmetry breaking by spatial confinement in tactoidal droplets of lyotropic chromonic liquid crystals. *Proceedings of the National Academy of Sciences* **108**, 5163–5168 (2011).
23. Sadoc, J.-F., Mosseri, R. & Selinger, J. V. Liquid crystal director fields in three-dimensional non-Euclidean geometries. *New J. Phys.* **22**, 093036 (2020).
24. Lavrentovich, O. D. Design of Chiral Domains by Surface Confinement of Liquid Crystals. *ACS Cent. Sci.* **6**, 1858–1861 (2020).
25. Mauguin, C. Sur les cristaux liquides de M. Lehmann. *Bulletin de Minéralogie* **34**, 71–117 (1911).
26. *Crystals That Flow: Classic Papers from the History of Liquid Crystals*. (CRC Press, 2004). doi:10.1201/9780203022658.
27. Zhang, S. S.-L., Phatak, C., Petford-Long, A. K. & Heinonen, O. G. Tailoring magnetic skyrmions by geometric confinement of magnetic structures. *Appl. Phys. Lett.* **111**, 242405 (2017).
28. Spethmann, J., Vedmedenko, E. Y., Wiesendanger, R., Kubetzka, A. & von Bergmann, K. Zero-field skyrmionic states and in-field edge-skyrmions induced by boundary tuning. *Commun Phys* **5**, 1–9 (2022).
29. Matsumoto, T. & Shibata, N. Confinement of Magnetic Skyrmions to Corrals of Artificial Surface Pits with Complex Geometries. *Frontiers in Physics* **9**, (2022).
30. Matsumoto, T. *et al.* Direct observation of $\Sigma 7$ domain boundary core structure in magnetic skyrmion lattice. *Science Advances* **2**, e1501280 (2016).
31. Shibata, K. *et al.* Large anisotropic deformation of skyrmions in strained crystal. *Nature Nanotech* **10**, 589–592 (2015).
32. Zázvorka, J. *et al.* Skyrmion Lattice Phases in Thin Film Multilayer. *Advanced Functional Materials* **30**, 2004037 (2020).
33. Kleemann, W. & Dec, J. Relaxor ferroelectrics and related superglasses. *Ferroelectrics* **553**, 1–7 (2019).
34. Parisi, G. Spin glasses and fragile glasses: Statics, dynamics, and complexity. *Proceedings of the National Academy of Sciences* **103**, 7948–7955 (2006).
35. Cowley, R. A. *et al.* The bicritical phase diagram of two-dimensional antiferromagnets with and without random fields. *Z. Physik B - Condensed Matter* **93**, 5–19 (1993).

Reviewers' Comments:

Reviewer #1:

Remarks to the Author:

The authors have satisfactorily addressed most of my recommendations. The revised manuscript and supplementary material now include more reproducible data and quantitative analysis, and the text has been improved for a general audience.

However, a weakness remains in that there are still no skyrmions observed in layers with a thickness below 110 nm. While the authors attribute this to the large unit cell of FCGT, it would be beneficial if they could provide information on how large the unit cell can be, as thicknesses as low as 10 -30 nm are still achievable. It would also be helpful if the authors could investigate the dependence of skyrmion size and structure on layer thickness. While 110-300 nm is still in the range of bulk behavior, demonstrating this dependence would strengthen the paper.

Overall, I recommend this paper for publication once this issue is addressed.

Reviewer #2:

Remarks to the Author:

I appreciate the authors' sincere response to my comments. The authors have addressed all my concerns and suitably revised the manuscript. Thus, I recommend the publication of the manuscript in Nature Communications.

Reviewer #3:

Remarks to the Author:

The authors have revised the manuscript according to the provided comments and suggestions. Several statements were changed and some paragraphs amended, which make the manuscript more accessible to readers. They provided point-by-point responses to all my questions. I am still of the opinion that the natural outlet for this type of work and results would be in another journal. However, this is not a decisive point for judgement. Therefore, I recommend the manuscript to be published in the current form.

Response to Reviewer Comments:

Reviewer #1 (Remarks to the Author):

The authors have satisfactorily addressed most of my recommendations. The revised manuscript and supplementary material now include more reproducible data and quantitative analysis, and the text has been improved for a general audience.

However, a weakness remains in that there are still no skyrmions observed in layers with a thickness below 110 nm. While the authors attribute this to the large unit cell of FCGT, it would be beneficial if they could provide information on how large the unit cell can be, as thicknesses as low as 10 -30 nm are still achievable. It would also be helpful if the authors could investigate the dependence of skyrmion size and structure on layer thickness. While 110-300 nm is still in the range of bulk behavior, demonstrating this dependence would strengthen the paper.

Overall, I recommend this paper for publication once this issue is addressed.

We thank the reviewer for the positive response.

Stabilization of magnetic skyrmions in thin nanoflakes has been identified as a core issue in magnetic layered materials, especially as dimensions approach the 2D limit. In systems hosting Bloch-type skyrmions, the size of the skyrmion is determined by the competition between Heisenberg exchange and the DMI and has been observed to be largely independent of sample thickness^{1,2}. In Néel-type systems, however, dipole-dipole interactions allow strong contributions to the energy of the system when the domain wall size is comparable to the thickness of the sample³. This means that Néel-type domains and skyrmions would generally be expected to follow Kittel scaling. In previous work⁴, we have rigorously characterized the condensation of the skyrmion phase as it changes with the thickness of the flake, where we observe that the skyrmion and cycloidal domain sizes follow Kittel's law.

Due to the anisotropy and DMI, thicker (several 100s of nm to bulk) flakes of FCGT will favor cycloidal stripe domains. At thicknesses below ~300 nm, application of a magnetic field causes these domains to irreversibly break apart and form the skyrmion phase. As shown in ⁴ and reproduced here, below ~100 nm, however, the pure skyrmion phase is no longer observed in the sample. We expect this to be due to the fact that in thin nanoflakes, when 2D effects dominate the energy landscape, the magnetization tends toward a single-domain, out-of-plane configuration. This has been previously indicated by the increasing magnetic hardness of the system with decreasing thickness⁵, which we can understand as an increasing out-of-plane anisotropy. Because of this, the intermediate skyrmion phase is not stable in thin (below ~100 nm) nanoflakes.

Figure R1 shows LTEM of a 60 nm thick nanoflake at room temperature where, with increasing magnetic field, we do not observe the breaking of the cycloidal domains into skyrmions, but rather just their dissolution into a single domain state. We can speculate that a pulsed current may help with the stabilization of skyrmions at these length scales, as has been observed in patterned metallic multilayers⁶, but that is beyond the scope of this work.

Figure 2 from ⁴, showing stable single phase skyrmions between ~ 125 - 250 nm thicknesses at zero field. After application of a finite magnetic field, the skyrmion can be stabilized in the nanoflakes above 100 nm. We also see that the size of both the Néel skyrmions and the cycloidal domains follow Kittel scaling.

Fig. R1 LTEM measurement in 60-nm-thick FCGT nanoflake at room temperature.

The following text has been added to the manuscript (P14L18):

“These thicknesses were chosen as they are both within the regime to stabilize Néel skyrmions as shown by previous work⁴.”

Reviewer #2 (Remarks to the Author):

I appreciate the authors' sincere response to my comments. The authors have addressed all my concerns and suitably revised the manuscript. Thus, I recommend the publication of the manuscript in Nature Communications.

We thank the reviewer for the helpful feedback and positive recommendation.

Reviewer #3 (Remarks to the Author):

The authors have revised the manuscript according to the provided comments and suggestions. Several statements were changed and some paragraphs amended, which make the manuscript more accessible to readers. They provided point-by-point responses to all my questions. I am still of the opinion that the natural outlet for this type of work and results would be in another journal. However, this is not a decisive point for judgement. Therefore, I recommend the manuscript to be published in the current form.

We thank the reviewer for the positive response.

References:

1. Park, H. S. *et al.* Observation of the magnetic flux and three-dimensional structure of skyrmion lattices by electron holography. *Nature Nanotech* **9**, 337–342 (2014).
2. Mühlbauer, S. *et al.* Skyrmion Lattice in a Chiral Magnet. *Science* **323**, 915–919 (2009).
3. Srivastava, A. K. *et al.* Observation of Robust Néel Skyrmions in Metallic PtMnGa. *Advanced Materials* **32**, 1904327 (2020).
4. Zhang, H. *et al.* Room-temperature skyrmion lattice in a layered magnet (Fe_{0.5}Co_{0.5})₅GeTe₂. *Science Advances* **8**, eabm7103 (2022).
5. Tan, C. *et al.* Hard magnetic properties in nanoflake van der Waals Fe₃GeTe₂. *Nat Commun* **9**, 1554 (2018).
6. Woo, S. *et al.* Observation of room-temperature magnetic skyrmions and their current-driven dynamics in ultrathin metallic ferromagnets. *Nature Mater* **15**, 501–506 (2016).